# Prevalence, Recognition, and Risk Factors of Constipation among Medically Hospitalized Patients: A Cohort Prospective Study

**DOI:** 10.3390/medicina59071347

**Published:** 2023-07-23

**Authors:** Jawahar Al Nou’mani, Abdullah M. Al Alawi, Juhaina Salim Al-Maqbali, Nahid Al Abri, Maryam Al Sabbri

**Affiliations:** 1Internal Medicine Residency Training Program, Oman Medical Specialty Board, Muscat 130, Oman; jhsm41191@gmail.com; 2Department of Medicine, Sultan Qaboos University Hospital, Muscat 123, Oman; 3Department of Pharmacy, Sultan Qaboos University Hospital, Muscat 123, Oman; jsmm14@gmail.com; 4Department of Pharmacology and Clinical Pharmacy, Sultan Qaboos University, Muscat 123, Oman; 5College of Medicine and Health Science, Sultan Qaboos University, Muscat 123, Oman; nahidalabri461@gmail.com (N.A.A.); msabbari98@gmail.com (M.A.S.)

**Keywords:** constipation, prevalence, risk factors, inpatient, in-hospital

## Abstract

*Background and Objective*: Constipation is a prevalent gastrointestinal condition that has a substantial impact on individuals and healthcare systems. This condition adversely affects health-related quality of life and leads to escalated healthcare expenses due to an increase in office visits, referrals to specialists, and hospital admission. This study aimed to evaluate the prevalence, recognition, risk factors, and course of constipation among hospitalized patients in medical wards. *Materials and Methods:* A prospective study was conducted, including all adult patients admitted to the General Medicine Unit between 1 February 2022 and 31 August 2022. Constipation was identified using the Constipation Assessment Scale (CAS), and relevant factors were extracted from the patients’ medical records. *Results:* Among the patients who met the inclusion criteria (n = 556), the prevalence of constipation was determined to be 55.6% (95% CI 52.8–58.4). Patients with constipation were found to be older (*p* < 0.01) and had higher frailty scores (*p* < 0.01). Logistic regression analysis revealed that heart failure (Odds ratio (OR) 2.1; 95% CI 1.2–3.7; *p* = 0.01), frailty score (OR 1.4; 95% CI 1.2–1.5; *p* < 0.01), and dihydropyridines calcium channel blockers (OR 1.8; 95% CI 1.2–2.8; *p* < 0.01) were independent risk factors for constipation. Furthermore, the medical team did not identify constipation in 217 patients (64.01%). *Conclusions:* Constipation is highly prevalent among medically hospitalized patients. To ensure timely recognition and treatment, it is essential to incorporate a daily constipation assessment scale into each patient’s medical records.

## 1. Introduction

Constipation is characterized by unsatisfactory bowel movements resulting from infrequent stools, difficulty passing stool, or a combination of both. Symptoms associated with constipation include, but are not limited to, hard stools, excessive straining, infrequent bowel movements, bloating, and abdominal pain. When these symptoms persist for more than one month, constipation is considered chronic [1]. The condition can arise idiopathically or as a result of various secondary causes, including neurological factors such as damage to the spinal cord, sacral parasympathetic nerves, or the impact of Parkinson’s disease on colon transit time. Other contributing factors may involve dyssynergic relaxation of the pelvic floor muscles, metabolic disorders such as hypothyroidism or diabetes mellitus, mechanical obstructing lesions like colorectal cancer, psychiatric disorders such as anorexia nervosa or irritable bowel syndrome, or medication side effects [2,3,4,5].

Constipation is a common gastrointestinal disease that significantly affects individuals and healthcare systems [6]. Constipation affects health-related quality of life and increases healthcare costs by increasing office visits, specialty referrals, and hospital admissions [7,8]. The reported prevalence of constipation among the community-dwelling adult population ranged from 2 to 35% [9,10,11,12]. The wide variation in reported prevalence may be due to differences in age groups, culture, diet, and environment as well as the differences in the definition of chronic constipation [6].

In contrast to community and outpatient settings, there has been limited research on the prevalence of constipation specifically among hospitalized patients. Previous studies have reported varying prevalence rates of constipation in acutely hospitalized patients, ranging from 14.8% to 65% [13,14,15,16]. The reported variation in prevalence rates can be attributed to the utilization of different definitions for constipation and the reliance on retrospective reviews of patients’ records [14]. Old age, use of certain medications such as opioids, and increased length of hospital stay were associated with increased prevalence of constipation in hospitalized patients [13,14,15,16].

In light of the limited availability of high-quality evidence regarding constipation in hospitalized patients, we conducted a prospective study with the aim of assessing the prevalence, recognition, and risk factors associated with constipation in medically hospitalized patients.

## 2. Materials and Methods

### 2.1. Study Setting, Design, and Population

This prospective cohort study was conducted at Sultan Qaboos University Hospital (SQUH), a multi-specialty tertiary hospital with 600 beds. SQUH is recognized as a prominent academic center and provides high-quality care to patients referred from all regions of the Sultanate of Oman [17,18,19]. The study encompassed all patients aged 12 years and above who were admitted under the care of General Internal Medicine (GIM) at SQUH between 1 February 2022 and 31 August 2022. Patients directly admitted to acute care units (the intensive care unit or high dependency unit), as well as those with colostomy or ileostomy, were excluded from the study. Informed consent was obtained from patients or their next of kin when necessary, such as in cases of lacking capacity due to conditions such as stroke, dementia, or others.

### 2.2. Sample Size

Based on previous studies, the reported prevalence of constipation in acutely hospitalized patients varied from 14.8% to 65% [13,14,15,16]. To determine the prevalence of constipation in our specific hospital setting, we estimated that a minimum of 391 patients would be required, considering a 95% confidence interval and a 5% margin of error.

### 2.3. Data Collection, Assessment Tools, and Definitions

Prospectively, a team of trained nursing trainees and medical students (acting as research assistants) collected the following data: demographic information, comorbidity details, medication records, admission data (length of stay, primary diagnosis), frailty scores, constipation-related information (prevalence, severity, treatment), and recognition of constipation by the treating team prior to the involvement of the research team (reflected in medical records or initiation of treatment). To assess constipation, the Constipation Assessment Scale (CAS) was utilized [20]. The constipation score for each hospitalized patient was determined at specific time points (days 0, 3, 7, 14, 21, 28) based on their length of stay. The primary diagnoses of the patients were categorized according to the 10th revision of the International Classification of Diseases (ICD-10).

Constipation was defined using CAS which consists of eight characteristics [20]. Each characteristic was rated on a three-point scale: ‘no problem’ (scored as 0), ‘some problem’ (scored as 1), and ‘severe problem’ (scored as 2). These characteristics include abdominal distension or bloating, changes in the amount of gas passed rectally, less frequent bowel movements, oozing liquid stool, rectal fullness or pressure, rectal pain with bowel movement, a small volume of stool, and the inability to pass stool. The total scores on the CAS range from 0 to 16. A score of 0 to 1 indicates no constipation, a score of 2 to 6 suggests mild to moderate constipation, and a score of 7 to 16 signifies severe constipation [21,22]. Previous validation studies demonstrated that CAS is a reliable, valid, and suitable scale to assess the presence and severity of constipation [20,23]. The highest score for each patient throughout admission was counted to ascertain the presence of constipation and assess the severity of constipation. In the case of terminally ill patients and non-communicating patients, information was obtained through clinical examination, specifically by observing for abdominal distention. Additionally, information was gathered from the patient’s attendant, such as the timing of the last bowel movement and the character of the stool.

The clinical frailty of patients was evaluated using a scale that comprised nine levels of dependency. These levels were defined as follows: 1—very fit, 2—well, 3—managing well, 4—vulnerable, 5—mildly frail, 6—moderately frail, 7—severely frail, 8—very severely frail, and 9—terminally ill. Based on the frailty score, the functional status of patients was categorized into three groups: independent (levels 1–4), partial dependence (levels 5–6), and full dependence (levels 7–9) [21,24].

All research assistants involved in data collection underwent a structured training program, which encompassed various aspects such as obtaining informed consent, proper data collection techniques, and administering the frailty scale and CAS accurately and appropriately.

### 2.4. Statistical Analysis

The Shapiro–Wilk test was used to assess the normality of continuous variables and expressed them as mean and standard deviation (SD) for normally distributed data or median and interquartile range (IQR) for non-normally distributed data. Differences between groups were assessed using one-way ANOVA for normally distributed variables or Kruskal–Wallis rank test for non-normally distributed variables. Categorical variables were reported as numbers, and percentages and differences between groups were compared using Pearson’s χ2 tests (or Fisher’s exact tests for expected cells < 5). All relevant factors with *p*-value < 0.3 were included in backward stepwise multiple regression analysis to identify independent predictors of constipation. Hazard ratios were calculated with 95% confidence intervals (Cis). Two-sided *p*-values < 0.05 were statistically significant. Statistical analysis was performed using the Stata v. 17.0 software package (StataCorp LLC, College Station, TX, USA).

## 3. Results

Out of all the patients screened, a total of 556 individuals met the inclusion criteria for this study. The median age of the participants was 58 years (IQR: 39–72), and 279 of them (50.18%) were male. Among the included patients, 339 individuals experienced constipation, with 309 patients had constipation on the first day of hospitalization and an additional 30 patients developing constipation after 72 h of hospitalization. This resulted in a calculated prevalence of constipation of 55.6% (95% CI 52.8–58.4) (Table 1).

Surprisingly, constipation was not recognized by the treating team in 217 of the constipated patients, accounting for 64.01% of the cases. The recognized cases were significantly associated with older age (*p* < 0.01) and higher CAS scores (*p* < 0.01).

Among patients diagnosed with constipation, 78.8% (n = 267) had mild to moderate constipation, while 21.2% (n = 72) experienced severe constipation. It was observed that patients with constipation tended to be older (*p* < 0.01) and had higher frailty scores (*p* < 0.01). The presence of certain comorbidities, including hypertension (*p* < 0.01), diabetes mellitus (*p* = 0.02), liver cirrhosis (*p* < 0.01), chronic kidney disease (*p* < 0.01), and heart failure (*p* < 0.01), as well as lower calcium levels (*p* = 0.01), lower hemoglobin levels (*p* < 0.01), higher serum creatinine levels (*p* = 0.02), and the use of opioids (*p* < 0.01), antihypertensives (*p* < 0.01), and dihydropyridines calcium channel blockers (CCB) (*p* < 0.01), were commonly associated with severe constipation (Table 2).

The results of the multivariate regression analysis revealed that preexisting heart failure (Odds Ratio (OR) 2.1; 95% CI 1.2–3.7; *p* = 0.01), higher frailty score (OR 1.4; 95% CI 1.2–1.5; *p* < 0.01), and the use of dihydropyridines calcium channel blockers (CCB) (OR 1.8; 95% CI 1.2–2.8; *p* < 0.01) were identified as independent predictors for constipation (Table 3).

Based on the primary diagnosis classified according to ICD-10, it was observed that diseases of the nervous system (*p* < 0.01) were more prevalent among patients without constipation. Conversely, respiratory system diseases were found to be more common in patients with constipation (*p* = 0.02) (Table 4).

## 4. Discussion

This is the first comprehensive prospective study with large sample size using a validated tool to assess constipation prevalence and risk factors in medically hospitalized patients, including those with COVID-19 infection. Constipation was highly prevalent (55.6%), yet significantly unrecognized by treating teams (64.01%). Heart failure, frailty score, and the use of dihydropyridines CCB emerged as independent predictors for constipation.

The prevalence of constipation among medically hospitalized patients was found to be 55.6% in this study. This figure exceeds the prevalence reported in a previous prospective cohort study, which reported a prevalence of 43% by day 3 of admission [13]. It is important to note that our study assessed patients for constipation over a longer duration, up to 28 days of hospitalization. In a recent retrospective cross-sectional study involving elderly hospitalized patients (n = 321), only 6% were diagnosed with constipation based on ICD-10 codes. However, clinical documentation indicated that 65% of these patients exhibited signs and symptoms of constipation [14]. Overall, the reported prevalence of constipation in acutely hospitalized patients varies widely, ranging from 14.8% to 65% [13,14,15,16]. These discrepancies can be attributed, in part, to differences in patient demographics, dietary factors, environmental conditions, and cultural influences, as well as variations in the definition of constipation and the reliance on retrospective record reviews, which may introduce inaccuracies [6,9,10,13,25].

In our cohort, patients with constipation were older (*p* < 0.01) and had higher frailty scores (*p* < 0.01). These findings are consistent with previous studies, where old age and increased friability were associated with an increased risk of constipation [26,27]. Old age and frailty are associated with decreased mobility, increased colonic transient time, poor nutrition, comorbidity, and polypharmacy, which could lead to increased risk of constipation [26]. Among individuals residing in the community, the prevalence of constipation tends to rise with age. In studies conducted among the population aged over 65 years, it was found that 26% of women and 16% of men identified themselves as experiencing constipation. Moreover, within a subgroup of patients aged 84 years, the proportion of individuals reporting constipation increased to 34% among women and 26% among men [1].

In contrast to the existing literature, our study revealed that patients with constipation had a lower prevalence of nervous system disorders [28,29,30]. Patients with nervous system disorders are less mobile; however, our constipated cohort has a lower percentage of fully dependent individuals. A community study on constipation showed there was a higher prevalence of constipation in more frail patients (41.7% frail vs. 33.9% pre-frail vs. 24.2% robust; *p* < 0.001), which is in line with our findings [31].

A higher prevalence of respiratory system diseases has been reported in the literature in patients with constipation. Constipation exposes patients to hypoxia and respiratory distress [32,33,34], along with the association with chronic obstructive pulmonary disease and asthma [33,35]. Constipation increases the risk of delirium in the elderly, which increases their risk of aspiration [36]. 

There were more patients with diabetes mellitus among the group of patients with constipation, especially those with severe constipation (*p* < 0.01). Previous studies linked diabetes mellitus to constipation, and it was reported as high as 60% prevalence of constipation in community-dwelling patients with diabetes mellitus [37,38]. The exact cause of the increased prevalence of constipation in patients with diabetes mellitus is unclear, but it may be related to diet, medications, or other comorbid conditions [37]. A recent study from Japan demonstrated that constipation in patients with type 2 diabetes mellitus is associated with diabetic retinopathy and diabetic nephropathy [38], and it was suggested that the presence of constipation in patients with diabetes mellitus might help in identifying patients who are at high risk of having progression of chronic kidney disease [39].

The study demonstrated that constipation is significantly associated with heart failure and liver cirrhosis. Patients with heart failure can develop thickened, edematous bowel if uncontrolled [40], and diuretic use can reduce body fluids leading to harder stool [41]. Furthermore, it was found that constipation prophylaxis in patients admitted with decompensated heart failure can shorten length of stay [42]. Previous studies have linked constipation to cardiovascular events. The alterations in the intestinal microbiota caused by constipation can contribute to the development of atherosclerosis, elevated blood pressure, and subsequent cardiovascular events. As constipation tends to increase with age, it frequently coexists with other cardiovascular risk factors. Furthermore, straining during bowel movements can lead to an increase in blood pressure, potentially triggering cardiovascular events such as arrhythmia, congestive heart failure, and acute coronary disease [41].

In patients with liver cirrhosis, constipation can be explained by several mechanisms, including delayed gastric emptying, prolonged small bowel transit time and gut edema induced by portal hypertension with gut vasculopathy or ascites [43].

We showed a significant association between the use of antihypertensive medications and constipation in our study. Specifically, dihydropyridines calcium channel blockers (CCBs), particularly amlodipine, were widely used by 70.9% of patients with hypertension in our cohort. Among the constipated group, 122 out of 339 (36%) patients were found to be taking dihydropyridines CCBs (*p* < 0.01). Previous reports showed that the prevalence of constipation associated with non-dihydropyridines CCBs like verapamil, ranging from 4% to 9% [44,45,46,47]. Additionally, higher doses of non-dihydropyridines CCBs have shown a stronger correlation with constipation. In contrast, amlodipine has been reported to cause constipation to a lesser extent (2%) [48,49]. Adverse drug reactions related to constipation sometimes lead to treatment discontinuation [50]. A recent study has demonstrated that amlodipine alone carries a four-fold higher relative risk of constipation incidents compared to its combination with atenolol [51]. Given that our constipated patients were elderly, caution is advised in combining constipation-inducing antihypertensive drugs (including ACE inhibitors) with amlodipine to prevent further complications [52].

Opioid-induced constipation is an established condition and was well described in the literature [53]. However, the use of opioids to improve constipation pain is widely discouraged. At the same time, it is a complex situation making it difficult to balance the urgency to treat pain, specifically in old patients. It is recommended to use other analgesics, assess for constipation pain and use laxatives [53,54].

Anemia is the most prevalent morbidity among the group with constipation. Lifestyle and dietary habits themselves can impact patients’ anemic status, while bowel habits as a result of food type, medications used to treat their anemia (e.g., oral iron supplement) or certain vitamins deficiency as vitamin B12 deficiency, which is linked to autonomic dysfunction, delayed gastric emptying, and constipation [55,56,57]. Hypocalcemia is associated with spasm and tetany, leading to autonomic manifestations including diaphoresis, bronchospasm, and biliary colic. Constipation is correlated to hypercalcemia rather than hypocalcemia [58]. However, it was reported that severe constipation improved after hypocalcemia correction [59].

This study has several strengths that contribute to its significance. Firstly, it addresses a research gap by focusing specifically on the prevalence, recognition, and risk factors associated with constipation in the hospital setting. Previous studies have primarily focused on community and outpatient settings. Secondly, the study utilized a prospective cohort design, allowing for the collection of data over an extended period of 28 days of hospitalization. This longer duration of assessment provides a more comprehensive understanding of constipation in hospitalized patients. Thirdly, the study employed a validated assessment tool, AS, to evaluate constipation prevalence and severity. The use of a validated tool enhances the reliability and validity of the study’s findings.

Despite its strengths, the study also has some limitations. One limitation is the exclusion of patients directly admitted to acute care units. Also, the study was conducted at a single tertiary hospital, which may limit the generalizability of the results to other healthcare settings.

## 5. Conclusions

This comprehensive prospective study on constipation in medically hospitalized patients provides valuable insights into the prevalence, recognition, and risk factors associated with constipation in this population. The study revealed a high prevalence of constipation (55.6%) among hospitalized patients, with constipation frequently going unrecognized by the treating teams (64.01% of cases). The study identified heart failure, frailty score, and the use of dihydropyridines calcium channel blockers (CCB) as independent predictors for constipation. These findings highlight the importance of considering constipation as a significant issue in hospitalized patients and emphasize the need for improved recognition and management by healthcare providers. Future research should aim to replicate these findings in diverse healthcare settings and explore interventions to improve the recognition and management of constipation in hospitalized patients, ultimately enhancing patient outcomes and reducing healthcare costs associated with constipation-related complications.

## Figures and Tables

**Table 1 medicina-59-01347-t001:** Demographics and clinical characteristics of the admitted patients (n = 556).

Characteristic, All (N = 556)	n (%) Unless Specified Otherwise
Age (IQR), years	58 (39–72)
Female	277 (49.82%)
Hypertension	282 (50.72%)
Diabetes Mellitus (DM)	226 (40.65%)
Liver Cirrhosis	15 (2.70%)
Chronic kidney disease (CKD)	73 (13.13%)
Heart failure (HF)	100 (17.99%)
Chronic obstructive pulmonary disease (COPD)	30 (5.4%)
Dementia	21 (3.78%)
Hypothyroidism	21 (8.54%)
Anemia	283 (51.55%)

**Table 2 medicina-59-01347-t002:** Patients’ demographics, clinical, and biochemical characteristics according to constipation.

Characteristic	All n = 556 (%) Unless Specified Otherwise	No Constipation n = 217 (%)	Mild–Moderate Constipation n = 267 (%)	Severe Constipation n = 72 (%)	*p* Value
Age (IQR), years	58 (39–72)	46 (29–65)	62 (44–74)	69.5 (55.5–76.5)	<0.01
Female	277 (49.82%)	103 (47.47%)	133 (49.81%)	41 (56.94%)	0.38
CAS’s highest score during hospitalization	2 (0–4)	0	3 (2–4)	7 (7–9)	<0.01
Comorbidities					
Hypertension	282 (50.72%)	87 (40.09%)	138 (51.69%)	57 (79.17%)	<0.01
Diabetes Mellitus (DM)	226 (40.65%)	81 (37.33%)	105 (39.33%)	40 (55.56%)	0.02
Liver Cirrhosis	15 (2.70%)	0	7 (2.62%)	8 (11.11%)	<0.01
Chronic kidney disease (CKD)	73 (13.13%)	18 (8.29%)	34 (12.73%)	21 (29.17%)	<0.01
Heart failure (HF)	100 (17.99%)	18 (8.29%)	58 (21.72%)	24 (33.33%)	<0.01
Chronic obstructive pulmonary disease (COPD)	30 (5.40%)	8 (3.69%)	19 (7.12%)	3 (4.17%)	0.22
Hypothyroidism	21 (8.54%)	3 (3.53%)	13 (10.66%)	5 (12.82%)	0.11
Depression	14 (2.55%)	4 (1.86%)	8 (3.03%)	2 (2.86%)	0.71
Dementia	21 (3.78%)	6 (2.76%)	12 (4.49%)	3 (4.17%)	0.60
History of abdominal surgery	23 (4.14%)	6 (2.76%)	13 (4.87%)	4 (5.56%)	0.42
History of gastrointestinal malignancy	11 (1.98%)	6 (2.76%)	5 (1.87%)	0	0.34
Frailty status					
Frailty score (IQR), the total number	4 (2–6)	2 (1–4)	5 (3–6)	6 (3–7)	<0.01
Independent	304 (54.68%)	165 (76.04%)	115 (43.07%)	24 (33.33%)	<0.01
Partial dependence	150 (26.98%)	28 (12.90%)	94 (35.21%)	28 (38.89%)	<0.01
Full dependence	102 (18.35%)	24 (11.06%)	58 (21.72%)	20 (27.78%)	<0.01
Biochemical profile					
Deranged liver function	243 (48.50%)	82 (41.84%)	123 (52.12%)	38 (55.07%)	0.05
Serum calcium (IQR), mmol/l	2.25 (2.14–2.36)	2.3 (2.18–2.38)	2.24 (2.14–2.34)	2.21 (2.09–2.32)	0.01
Serum potassium (IQR), mmol/l	4.2 (3.9–4.6)	4.2 (3.9–4.6)	4.2 (3.9–4.6)	4.3 (3.8–4.6)	0.05
Hemoglobulin (IQR)	11.1 (9.2–13)	12.3 (9.8–13.5)	10.8 (9.1–12.7)	9.85 (8.6–11.9)	<0.01
Serum creatinine (IQR), mmol/l	75 (57–108.5)	71 (57.5–102)	75.5 (55.5–109)	93 (63–150.5)	0.02
TSH, (IQR), mmol/l	2.07 (1.16–3.49)	1.96 (1.03–3.53)	1.93 (1.16–3.47)	2.52 (1.61–3.51)	0.35
T4 (IQR), mmol/l	15.8 (13.9–18.4)	16 (14.2–17.9)	15.6 (13.7–18.6)	16.7 (12.8–18.5)	0.35
Medications					
Antidepressants	4 (0.72%)	3 (1.40%)	1 (0.38%)	0	0.31
Antipsychotic medications	17 (3.08%)	5 (2.33%)	8 (3.02%)	4 (5.56%)	0.39
Iron supplements	49 (8.88%)	16 (7.44%)	29 (10.94%)	4 (5.56%)	0.23
Opioids	101 (18.30%)	32 (14.88%)	46 (17.36%)	23 (31.94%)	<0.01
5-HT3antagonists	116 (21.01%)	51 (23.72%)	49 (18.49%)	16 (22.22%)	0.36
Antihistamines	72 (13.04%)	28 (13.02%)	35 (13.21%)	9 (12.50%)	0.99
Antihypertensives	230 (41.67%)	68 (31.63%)	110 (41.51%)	52 (72.22%)	<0.01
Dihydropyridines Calcium Channel Blockers (CCB)	163 (29.53%)	41 (19.07%)	93 (35.09%)	29 (40.28%)	<0.01

**Table 3 medicina-59-01347-t003:** Risk factors multi-regression analysis: independent risk factors.

Overall Constipation	Odds Ratio	95% CI	*p* Value
Heart failure	2.09	1.18–3.73	0.01
Frailty	1.35	1.24–1.48	<0.01
CCB	1.78	1.26–2.76	<0.01

**Table 4 medicina-59-01347-t004:** Primary diagnosis classified according to ICD-10.

Primary Diagnosis (ICD-10)	All (n = 556)	No Constipation n = 217 (%)	Mild–Moderate Constipation n = 267 (%)	Severe Constipation n = 72 (%)	*p* Value
Infectious disease (A00–B99)	46 (8.27%)	24 (11.06%)	19 (7.12%)	3 (4.17%)	0.14
Hematological diseases (D50–D89)	18 (3.24%)	6 (2.76%)	11 (4.12%)	1 (1.39%)	0.49
Endocrine, nutritional, and metabolic diseases (E00–E90)	51 (9.17%)	17 (7.83%)	24 (8.99%)	10 (13.89%)	0.30
Mental and behavioral disorders (F00–F99)	16 (2.88%)	9 (4.15%)	6 (2.25%)	1 (1.39%)	0.43
Diseases of the nervous system (G00–G99)	36 (6.47%)	24 (11.06%)	12 (4.49%)	0	<0.01
Diseases of the circulatory system (I00–I99)	94 (16.91%)	29 (13.36%)	49 (18.35%)	16 (22.22%)	0.15
Diseases of the respiratory system (J00–J99)	91 (16.37%)	29 (13.36%)	42 (15.73%)	20 (27.78%)	0.02
Diseases of the digestive system (K00–K93)	52 (9.35%)	18 (8.29%)	28 (10.49%)	6 (8.33%)	0.68
Diseases of the genitourinary system (N00–N99)	54 (9.71%)	14 (6.45%)	34 (12.73%)	6 (8.33%)	0.10
Others	98 (556%)	47 (21.66%)	42 (15.73%)	9 (12.50%)	0.19

## Data Availability

The data presented in this study are available on request from the corresponding author. The data are not publicly available due to patients related privacy issues.

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
