# Peer review of "Prevalence, Recognition, and Risk Factors of Constipation among Medically Hospitalized Patients: A Cohort Prospective Study"

_medicina, 2023, doi:10.3390/medicina59071347_

Round 1

Reviewer 1 Report

Constipation is often an overlooked aspect of patient care, can prolong hospital stay, increase financial cost and add to staff nursing care time. The topic is therefore interesting, but certainly not new, as can be found in various reviews, among which we could mention, not reported in the text

Dupont et al. Anatomy, physiology, and updates on the clinical management of constipation. Clin Anat 2020, 33(8):1181-1186.

Scott SM et al. Chronic constipation in adults: Contemporary perspectives and clinical challenges. 1: Epidemiology, diagnosis, clinical associations, pathophysiology and investigation. Neurogastroenterol Motil 2021, 33:e14050.

The merit of this work is to have used a validated tool to assess constipation and to have conducted a prospective study. The authors highlight a high prevalence of constipation in hospitalized patients, highlighting many frailties. even if this conclusion would seem rather obvious.

The bibliography must be written in a more uniform manner.

Author Response

Authors response:

Thank you for the remark . We have added both references as per the suggestion .Also, we have revised the bibliography and re-written them in uniform manner .

Reviewer 2 Report

In this study, authors investigated constipation, a prevalent gastrointestinal condition. All adult patients admitted to the General Medicine Unit was conducted in this prospective study. Among the patients, the prevalence of constipation was determined to be 55.6%. Patients with constipation were found to be older and had higher frailty scores. Heart failure, frailty score, and dihydropyridines-calcium channel blockers were independent risk factors for constipation. Finally, they made a conclusion that constipation is highly prevalent among medically hospitalized patients. To ensure timely recognition and treatment, it is essential to incorporate a daily constipation assessment scale into each patient's medical records. In general, this study is very interesting and here are some comments from this reviewer:

1. The tables of this manuscript should be well organized.

2. The pathogenesis of constipation should be introduced.

Author Response

  1. The tables of this manuscript should be well organized.

Authors response:

We have revised all tables as per the suggestion

  1. The pathogenesis of constipation should be introduced.

Authors response:

We have added section addressing the pathogenesis of constipation.

“Constipation is characterized by unsatisfactory bowel movements resulting from infrequent stools, difficulty passing stool, or a combination of both. Symptoms associated with constipation include, but are not limited to, hard stools, excessive straining, infrequent bowel movements, bloating, and abdominal pain. When these symptoms persist for more than one month, constipation is considered chronic [1]. The condition can arise idiopathically or as a result of various secondary causes, including neurological factors such as damage to the spinal cord, sacral parasympathetic nerves, or the impact of Parkinson's disease on colon transit time. Other contributing factors may involve dyssynergic relaxation of the pelvic floor muscles, metabolic disorders such as hypothyroidism or diabetes mellitus, mechanical obstructing lesions like colorectal cancer, psychiatric disorders such as anorexia nervosa or irritable bowel syndrome, or medication side effects [2-5]”